# Microanatomical Changes in the Leaves of *Arundo donax* (L.) Caused by Potentially Toxic Elements from Municipal Sewage Sediment

**DOI:** 10.3390/plants13050740

**Published:** 2024-03-06

**Authors:** Csilla Tóth, László Simon, Brigitta Tóth

**Affiliations:** 1Department of Agricultural Sciences and Environmental Management, Institute of Engineering and Agricultural Sciences, University of Nyíregyháza, Sóstói Str. 31/b, H-4400 Nyíregyháza, Hungary; toth.csilla@nye.hu (C.T.); simon.laszlo@nye.hu (L.S.); 2Institute of Food Science, Faculty of Agricultural and Food Sciences and Environmental Management, University of Debrecen, Böszörményi Str. 138, H-4032 Debrecen, Hungary

**Keywords:** *Arundo donax*, leaf, microanatomy, municipal sewage sediment, potentially toxic elements

## Abstract

An open-field 3-year-long microplot experiment was set up with three micropropagated lines (SC Blossom, BFT Indiana, and STM Hajdúsági) of giant reed (*Arundo donax* L.). Plants were grown on a soil cover of a former sewage settling pond located in Debrecen Lovász-Zug, Hungary. Soil cover of the sewage sediment was moderately contaminated with various toxic elements (As, Ba, Cd, Cr, Cu, Mn, Ni, Pb, and Zn). The highest total concentration of examined toxic elements in leaves was found in the BFT Indiana line (∑326 mg/kg), while in the SC Blossom and STM Hajdúsági lines, ∑210 mg/kg and ∑182 mg/kg were measured, respectively. The highest Zn concentration (117 mg/kg) was found in the leaves of in BFT Indiana line and was 67% higher than that in SC Blossom and 95% more than in the STM Hajdúsági line. The BFT Indiana leaves showed typical signs of adaptation to heavy metal stress in the case of numerous micromorphometric characteristics. The extent of leaf mesophylls decreased, and the number of bulliform cells and phytoliths, as well as the sclerenchymatous stock, increased. The size of the vascular bundles was reduced. The size of the stomata decreased while the stomatal density increased. It can be concluded that the BFT Indiana line had the best adaptational response to heavy metal stress.

## 1. Introduction

Giant reed (*Arundo donax* L., *Poaceae*) is a robust, perennial, rhizomatous grass with C3-type photosynthesis. This plant is utilized in different parts of the world for industrial, construction, agricultural, environmental, and bioenergy purposes [1]. Because its aboveground organs can produce annually 10–20 dry tons of biomass per hectare without irrigation, this plant is a perspective energy crop. Under optimal growth conditions, the aboveground biomass yield of giant reed can exceed 35 dry tons annually [1,2]. Arundo biomass is suitable for burning; pyrolysis; ethanol, biogas, and silage production; and forage feedstock production [3,4,5,6].

This species tolerates a wide range of soil and climatic conditions and can be cultivated on marginal and degraded lands [1]. Previous studies showed that *Arundo*, besides high productivity, has tolerance to water or soil trace element exposure. This plant species is therefore also a candidate for phytomanaging water and soils contaminated by trace elements [7,8,9,10,11,12,13].

Giant reed, as an energy plant, can be grown on the same plot for up to 10–15 years without replanting. With an adequate nutrient supply, this period can be extended, but the yield gradually decreases [2]. The aboveground biomass yield of *Arundo* can be stimulated by soil application of various mineral or organic fertilizers, additives, and by-products, including biosolids (e.g., municipal sewage sludge) and biomass ash [2,14]. However, the application of various soil amendments (e.g., municipal sewage sludge or ash) can enhance, not only the uptake rate of beneficial elements (e.g., nitrogen or potassium) but also the rate of potentially toxic element (PTE) accumulation in the giant reed organs [14,15,16]. This may have an impact on the toxic metal concentration of harvested shoots, and on the toxic metal concentration of ash after biomass burning. Since *Arundo* can also be grown as a fodder material [17], its shoots can accumulate PTEs from the contaminated growth media, and PTEs can enter the food chain via fodder-consuming farm animals.

In our preliminary open-field micro plot experiments [6,7] with two ecotypes of *Arundo*, the interaction with two types of biowaste (municipal sewage sludge compost—MSSC; municipal green waste compost—MGWC) was studied. It was found that even from high provocative doses (51 dry tons/ha) of MSSC (slightly contaminated with PTEs) added to the soil, the accumulation of Cd or Pb was negligible in the shoots of *Arundo*. The most important microanatomical parameters in *Arundo* leaves were also investigated during these experiments [18]. It was found that the soil-applied MSSC or MGWC reduced the thickness of the epidermis. Treatments expanded the width of vein islands between the parallel veins and the number of epidermis cell lines of the vein islands. The soil application of the above biowastes caused a significant increase in stoma numbers. It can be supposed that the higher stoma number enables a higher rate of CO_2_ accumulation, which supports a higher biomass production of *Arundo*.

Guo and Miao [19] conducted a greenhouse experiment to elucidate the growth changes and tissue anatomical characteristics in *Arundo*, which was cultivated in soils contaminated with As, Cd, and Pb. The results showed that giant reed rapidly grows with considerable biomass of shoots in contaminated soil, possessing strong metal tolerance with limited metal translocation from roots to shoots. When As, Cd, and Pb concentrations in the soil exceeded 334, 101, and 2052 mg/kg, respectively, stem microanatomical images became heterogeneous and the secretion in vascular bundles increased significantly.

Plants have numerous strategies to moderate the negative effects of heavy metal stress. To protect their photosynthetic tissues, a crucial step in their strategy is to prevent the translocation of heavy metals and toxic elements into the leaves [20,21]. Although only a moderate amount of metals translocate into the leaves, their presence, even in small doses, can cause serious microanatomical changes (decrease in the size of parenchymal cells, decrease in the extent of cellular elements of the xylem, decrease in the size of the stomata, decrease in the amount of photosynthetic pigments) [22].

Several authors [23,24,25] describe a decrease in the size of the stomata and their closure due to long-term metal exposure. All of these changes have a negative effect on transpiration, photosynthesis, and gas exchange.

The changes in the size and number of stomata are a manifestation of the plants’ response to environmental changes, and they play an important role in regulating the absorption of pollutants by plants [26]. Several authors describe the increase of the number of stomata as a result of heavy metal, for example, Cd exposure [22,25,27,28,29,30,31]; Pb, Zn, and Cu are also described to have similar effects [32,33,34]. However, others describe the opposite. For example, Kasim [35] reported that, in the case of *Sorghum bicolor*, the stomatal density decreased as a result of Cd exposure. However, many authors [26,36,37,38,39,40] also described an increase in the stomatal density parallel to the decrease in the size of the stomata. According to them, this adaptation is considered beneficial because by increasing the number of stomata, the plants can provide CO_2_ for their photosynthesis, and they can do so without excessive water loss by decreasing the size of the pores.

Wainwright and Woolhouse [41] also describe a decrease in the size of the stoma apparates alongside an increase in the stomatal density. They established that the size decrease in the subsidiary cells, which is a result of metal toxicity, has an important role in the decrease in the stoma apparates. Numerous studies show that the size of the subsidiary cells is correlated with the degree of the heavy metal pollution; Cd contamination causes a decrease in the length of the subsidiary cells [42], Pb has a negative effect on the width of the subsidiary cells [43], Cu disrupts the arrangement of subsidiary cells around guard cells [44,45], and Pb, Al, and Cd can be also responsible for the disruption of the guard cells [46,47]. Other studies also describe that Cd, Cu, and As can cause morphological and structural changes in the guard cells [44,45,47].

Several authors [48,49,50] establish that in the case of herbaceous plants grown on soil contaminated with highly toxic elements, with a rise in the degree of contamination and the concentration of toxic elements in the plant organs, the thickness of the leaf lamina significantly decreases, and the extent of the intercellular spaces and the organization of the cells change [26,51].

The thickness of the sclerenchyma bridges increases as a result of heavy metal exposure both on the abaxial and adaxial sides. The heavy metals practically bind to the cell walls of the sclerenchymatous tissues. Because of this, their translocation into the leaf and their access to assimilating tissues is obstructed. This is also a great adaptation strategy to reduce the heavy metal stress of photosynthetically active tissues. Vollenweider et al. [52] report a similar conclusion. They describe that the cells of the increased collenchyma in the leaves can accumulate a significant amount of heavy metal ions in their cell walls, which minimizes the possibility of these heavy metals accessing the photosynthetic tissues and damaging them.

In the case of species belonging to the *Poaceae* family, phytoliths can be good indicators of the increasing concentration of toxic elements in soils. Phytoliths are three-dimensional bodies formed from hydrated amorphous silicon dioxide and are particularly widespread in the *Poaceae* family [53,54]. Plants take up silicon in the form of monosilicic acid (H_4_SiO_4_) from the soil. After it has polymerized to amorphous silicon dioxide, it is either stored inside the cells (vacuoles) or in intercellular spaces [55]. The stored silicon dioxide has a role in numerous biological processes, including the insurance of mechanical strength, protection against diseases, and mitigation of abiotic stress (metal toxicity, salt, drought, and heat stress) [56,57]. Phytoliths can be found in numerous parts of the plants. They can be found in the root cells, the epidermal cells of the stem, the bracts that protect the flower and inflorescence, and the leaves. In leaves, they play a role in the regulation of evaporation, and the phytoliths appearing in the form of bulliform cells reduce the plant surface area exposed to evaporation [55].

Considering the above preliminaries, we aimed to investigate the uptake or accumulation of nine selected PTEs (arsenic, As; barium, Ba; cadmium, Cd; chromium, Cr; copper, Cu; manganese, Mn; nickel, Ni; lead, Pb; and zinc, Zn) in the leaves of three giant reed (*Arundo donax* L.) lines grown in a microplot open-field experiment. It was assumed that during the three growth periods (years 2018, 2019, and 2020), PTEs from municipal sewage sediment (MSS) would accumulate in the leaves of giant reeds, and this would differentially influence the microanatomy parameters there.

We aimed to examine three *Arundo donax* lines and select the one that is the most suitable for the recultivation of areas contaminated with toxic elements such as our sample area. This line should be able to accumulate a large amount of toxic elements while having good histological adaptation properties to toxic elements and exceptional biomass yield. In order to do this, we performed comparative leaf anatomy analysis of three *Arundo* lines from soil contaminated with toxic elements.

## 2. Results and Discussion

### 2.1. Potentially Toxic Elements in Experimental Soil and Arundo Leaves

The basic characteristics of the soil covering the MSS were the following at 0–30 cm depth: loamy texture; pH-H_2_O: 7.67–7.81; pH-KCl: 7.28–7.34; total salt (m m^−1^ %): 0.055–0.058; CaCO_3_ (m m^−1^ %): 2.13–2.45; humus (m m^−1^ %): 2.16–2.39; CEC (cmolc kg^−1^): 25.1–25.7. The average (n = 4) macronutrient concentrations were the following: P-1122, K-1859, Ca-17921, Mg-5055 mg kg^−1^, as determined from cc. HNO_3_–cc. H_2_O_2_ extract, with analysis performed following the instructions of the Hungarian Standard MSZ 21470-50 2006 [58].

Table 1 presents the “pseudo-total” concentrations of potentially toxic elements (PTEs) in the topsoil (0–25 cm) of three experimental sites before starting the open-field experiment with Italian reed. From the data, it is obvious that uneven covering of MSS and its uneven mixing with cover soil resulted in uneven PTE contamination of the topsoil. The lowest PTE concentrations were found in Location 1, where the STM giant reed line was grown, as compared to Location 2, where the BFT line was cultivated, and Location 3, where the SC Blossom line was cultivated. The largest differences in the PTE concentrations were observed for Zn and Pb; at Location 2 or 3 the measured values were two or three times higher than in Location 1. Ba, Cr, Cu, and Zn concentrations measured in the topsoil of Location 2 or 3 were higher, while As, Cd, and Ni concentrations were lower than the Hungarian threshold limits (As–15, Ba–250, Cd–1, Cr–75, Cu–75, Ni–40, Pb–100, and Zn–200 mg kg^−1^) for soil pollution [59]. In Location 1, only the Cr concentration (126 mg kg^−1^) exceeded the Cr–75 mg kg^−1^ pollution threshold.

Table 2 demonstrates the concentrations of PTEs in the leaves of Italian reeds during the autumn of 2018 or 2019 or during the summer of 2020. Since *Arundo* is a perennial plant (its dry shoots were cut off in each March of the experiment), it was assumed that the PTE accumulation would change with the aging of the plants, and this would influence the microanatomical parameters of leaves. In general, the concentrations of accumulated PTEs in leaves reflect the amounts of PTEs in the soil. The measured As, Ba, Ni, and Pb concentrations in leaves are low and are definitively lower than the concentrations supposed to be excessive or toxic in mature leaf tissue, generalized for various plant species [7,60]. Generally, 0.05–0.2 mg kg^−1^ Cd is a normal range in the leaves of various plant species, while 5–30 mg kg^−1^ is excessive or toxic [60,61]. Cd concentrations in Arundo leaves varied between 0.191 and 0.634 mg kg^−1^ in the three sampling years (Table 2). The highest Cd concentrations (0.467–0.634 mg kg^−1^, Table 2) were found in the leaves of 113-week-old Arundo plants on 25 June 2020. At this sampling time, the new shoots of the plants were 7 weeks old, since the dried, withered Italian reed shoots were cut off at the end of March 2020. Regarding Cu, 5–30 mg kg^−1^ is sufficient or normal, while 20–100 mg kg^−1^ is excessive or toxic for plants [60,61]. The values of Cu in the leaves of Arundo plants (Table 2) were in the normal range at all sampling times. In the leaves of various plant species, 27–150 mg kg^−1^ Zn is sufficient or normal, and 100–400 mg kg^−1^ Zn is excessive or toxic [60,61]. The highest Zn concentrations (147–187 mg kg^−1^, Table 2) were found in the leaves of young 16-week-old Arundo plants on 13 September 2018.

Soil sampling was repeated at the end of the experiment. At that time, soil samples were collected not only from the topsoil (0–30 cm) but also from the 30–60 cm subsoil layer. Besides the “pseudo-total”, the “plant-available” concentrations of the PTEs were also determined in soil extracts. Results are shown in Table 3.

Results from 2021 (Table 3) confirm our observations from year 2018 (Table 1) that PTEs were present in the topsoil covering the MSS. From the data, it is again obvious that uneven soil mixing with MSS resulted in uneven PTE contamination of the topsoil. Similarly to the year 2018, the lowest “pseudo-total” PTE concentrations were found in Location 1 (where the STM giant reed line was grown), and the highest PSE concentrations were found in Location 3 (where the SC Blossom line was cultivated). As a general trend, in the subsoil at the 30–60 cm layer, higher amounts of “pseudo-total” or “plant-available” PTEs were found than in the topsoil. These enhancements can be attributed to various rates of downward migration (leaching) of PTEs from the topsoil, to the upward migration of certain PTEs from the MSS to the subsoil, or to the transfer of certain PTEs from the topsoil to the Italian reed shoots (Table 2).

The bioconcentration factor (BCF) is a quotient between element concentrations in plants/soil. BCFs (Appendix A) calculated from September 2018 values of accumulated PTEs in the leaves of various lines of Arundo (Table 2) and from May 2018 “pseudo-total” concentrations of PTEs in 0–25 cm deep soils (Table 3) were in this order: Cd > Zn > Mn > Cu > Ni > Pb > As > Ba > Cr. The highest BCFs were found for Cd (1.093) and Zn (1.090) in the STM line. Generally, the STM line had higher BCFs for other PTEs than the BFT or SC Blossom line. This tendency was confirmed when BCFs were calculated by dividing the concentrations of PTEs in leaves collected in June 2020 by the “pseudo-total” concentrations of PTEs in 0–30 cm deep soil samples from July 2021 (Appendix A). In that case, the order of BCFs was Cd > Zn > Mn > Cu > Ni > As > Ba > Cr > Pb, and again the STM Arundo line had the highest bioconcentration of PTEs in leaves. Based on subsoil data (30–60 cm soil depth, “pseudo-total” PTE concentrations) from the above sampling periods, the BCF order was Cd > Zn = Mn > Cu > Ni > Ba > As > Cr > Pb. In this case, however, the BFT Arundo line showed the highest BCF values (Appendix A). Based on the above data, it can be concluded that the elements Cd, Zn, Mn, Cu, and Ni were mostly bioconcentrated from the soil in the leaves and that these elements are assumed to have caused micromorphological changes in the leaves.

According to Buscaroli [62], an element’s pseudo-total and available fractions are preferable when assessing the element bioconcentration ability of plants. Therefore, we calculated BCFs as the ratio of the PTE concentration in the Arundo leaves not only to the PTE “pseudo-total” concentration but also to the “plant-available” concentration in the soil (Appendix A). BCFs calculated from June 2020 values of PTEs in the leaves of various lines of Arundo (Table 2) and from July 2021 “plant-available” concentrations of PTEs in soil from 0 to 30 cm in depth (Table 3) were in this order: Cr > Cd > Zn > Ni > Cu > Mn > Ba > As > Pb, while for soils from 30 to 60 cm in depth, the order was the following: Cr > Ni > Cd > Zn > Mn > Cu > Ba > As > Pb (Appendix A).

Trace element concentrations in plants reflect, in most cases, their abundance in the growth media. Generally, Cr is very slightly soluble in soil solution and is not easily taken up by plants; As and Pb are relatively strongly adsorbed to soil particles and are not readily transported to aboveground parts of plants; Cu, Mn, and Ni are mobile in soil and readily taken up by plants; and Cd and Zn are very mobile in soil and are easily bioaccumulated by plants [60,61]. These general observations are largely supported by our calculated BCFs (Appendix A) for Arundo leaves and by measured PTE concentrations in the leaves (Table 2).

A correlation analysis was performed between the PTE concentrations measured in the leaves of various Arundo lines and the PTE concentrations found at the different depths (topsoil, subsoil) and in the “pseudo-total” or “plant-available” fractions of the soil (Appendix A). In most cases, we did not find a strong correlation between these factors. When examining correlation coefficients (r) greater than 0.8, positive or negative correlations between leaf and soil PTE concentrations were generally observed mostly for STM and BFT lines. Besides the calculation of correlation coefficients with the formula used in Microsoft Excel Office 2016 software, the significance of the correlation was also checked with a two-tailed Pearson correlation utilizing IBM SPSS Statistics 26.0. A positive correlation (r = 0.972) between Cd concentration measured during June of 2020 in STM Arundo line leaves and the “pseudo-total” Cd concentration measured in the topsoil during July of 2021 proved to be significant at the 0.05 level. A negative correlation (r = −0.984) between Cd concentration measured during June of 2020 in BFT line leaves and the “plant-available” Cd concentration measured in the topsoil during July of 2021 also proved to be significant at the 0.05 level. A positive correlation (r = 0.977) between the Pb concentration measured during June of 2020 in BFT line leaves and the “pseudo-total” Pb concentration measured in the subsoil during July of 2021 proved to be significant at the 0.05 level. In the SC Blossom line, in the case of Ba, the positive correlation (r = 0.986) between the concentration of this PTE in leaves sampled in June of 2020 and in subsoil sampled in July of 2021 proved statistically significant at the 0.05 level.

### 2.2. Anatomical Responses of the Leaf

The leaf mesophyll of Arundo is isolateral and homogenous; its anatomical characteristics suggest that giant reed prefers moist and shady habitats. The veins stand out from the leaf blade, surrounded by a parenchymatous bundle sheath, which sharply separates the vascular bundles from the spongy parenchyma. The vascular bundles are connected to the epidermis by sclerenchyma fibers (Figure 1).

Around the vascular bundles, partial Kranz anatomy can be observed; this may play a role in the efficient photosynthesis of the giant reed, which is much more intense than that of all C3 plants [8,63]. Due to this characteristic of the giant reed, it can produce a high level of biomass, which greatly exceeds the biomass production of C4 photosynthetic plants [2,64,65].

A thick cuticle layer covers the surface of the epidermis. The epidermis is made out of long and short cells, which is typical of the *Poaceae* family [66,67]. Characteristic bulliform cells can be observed among the adaxial epidermis cells. These cells have no pigment, are thin-walled, have wide cavities, and are wedged deep between the cells of the leaf mesophylls. The leaves of the giant reed are amphistomatic and have paracytic stomatal complexes, which is typical of the *Poaceae* family. The stomatal complexes are mesomorphic, consisting of two dumbbell-shaped guard cells and two dome-shaped subsidiary cells, and their position is lateral (Figure 2). Stomata can be seen between the epidermal cells, in the intercostal regions, and, in the case of the adaxial epidermis, directly on both sides of the bulliform cells. Their density is higher on the abaxial surface than on the adaxial surface [68].

There are many differences between the micromorphometric characteristics of the adaxial and abaxial epidermis, such as the number of cell rows in the rib zone, the size of long cells (length, width), and the frequency of phytoliths (silica bodies). The costal zones are narrower on the adaxial epidermis, and the anticlinal wall of their long cells is straight on the adaxial epidermis and wavy on the abaxial epidermis. Long cells on the adaxial epidermis are on average longer than those on the abaxial epidermis. There are no papillae on the long cells, and trichomes arise from the short cells. In the part above and between the veins, phytoliths can be found on one hand as silicified epidermal long cells (elongate spiny with concave end, elongate spiny with pavement, elongate sinuous), bulliform cells/bulliform phytoliths (their lumens are filled with silicon dioxide), epidermal short-celled phytoliths (bilobates, bilobate with nodular shank, cross, trapezoid, microhairs), as well as a guard and subsidiary cells with silicified cell walls [57,69,70].

The amount of silica that accumulates is influenced by the following: (1) extrinsic, abiotic environmental factors, such as temperature, humidity, soil type and soil moisture, toxic elements, and the abiotic stress caused by them and (2) intrinsic factors, such as the phenological state of the plant [57,70,71,72].

Due to the increasing accumulation of heavy metals and toxic elements in the leaves, we detected several changes in the microanatomical parameters of the leaves of the Arundo lines (Figure 3). Similar to the observations of other authors [21,35,40,41,60,63,73,74] regarding species belonging to the *Poaceae* family, it was observed that with the toxic element load, the thickness of the epidermis and the extent of the leaf mesophyll decreased.

In the case of all lines examined over the years, we can establish that both the adaxial and abaxial epidermis showed a decreasing tendency in thickness (Table 4). The only exception was the BTF line, where the adaxial epidermis did not show any changes in its extent over the years. In the case of BTF and STM lines, the decrease was significant in the case of the abaxial epidermis. However, in the case of the SC Blossom line, there was only a statistically verifiable decrease in the thickness of the adaxial epidermis.

When the leaves of the plants are exposed to heavy metal contamination, the collapse of the cells building the mesophyll is observable; therefore, the extent of the mesophyll inside the leaf lamina decreases [50,75]. A similar tendency was observable in the case of all three lines: compared to that in the first year of examination (2018), the extent of the mesophyll decreased in the following years (Table 4).

In the case of the BTF and STM lines, the extent of the mesophyll inside the leaf laminae showed a significant decrease over the examination years, while in the case of the SC Blossom line, we could only establish a statistically verified difference in the first two years (Figure 3). A similar phenomenon has been observed by several authors, such as Sridhar et al. [76] in barley (grown with different levels of cadmium and zinc) and Santana et al. [77] in *Setaria parviflora* and *Paspalum urvillei* grown via hydroponics under different concentrations of iron. Our observations are very similar and support the observations of Koleva et al. [78]: as a result of their experiment with durum wheat, they found that heavy metals, mainly Zn^2+^, reduced the mesophyll thickness. Müller et al. [79] and Brandão et al. [80] also observed the collapse of epidermal cells parallel with mesophyll cells, resulting in a reduced leaf blade thickness caused by heavy metal contamination.

After examining the extent of the vascular bundles, it can be established that over the three examination years, their extent showed a significant decrease in the case of the BTF line. The SC Blossom and STM lines only showed a significant decrease between the first two years (Table 4). The reduction in the elements of the vascular tissues is in clear connection with the rising heavy metal load [21,80,81,82].

Gomes et al. [21] concluded that the thickened adaxial and abaxial epidermis and the enlarged bulliform cells could be a strategy that aims to minimize water loss through transpiration. The decrease in evaporation and transpiration flow reduces the translocation of heavy metals from the roots to the leaves. Thanks to the leaf curling caused by bulliform cells, the evaporation surface of the leaf decreases. This way, a more humid environment and a wetter microclimate can develop around the stomata [83]. This is also an adaptation strategy that aims to prevent the heavy metals from accessing the leaves [81,82].

In the case of all lines examined, the number of bulliform cells increased, and the differences between the examined years were significant. However, the width of the bulliform cells showed a decreasing tendency over the years. This decrease was significant in the case of the BTF and STM lines. The total height of the bulliform cells decreased in every line, but in the case of the BTF and SC Blossom lines, the decrease only showed a significant difference in the first two examination years (Table 4, Figure 3). The above observations are consistent with the observations of many other authors [18,21,40,41,74,83,84], which established that the number of bulliform cells showed an increasing tendency; however, their extent decreased. Contrary to the above, Melo et al. [40] observed that the extent (width and height) of the bulliform cells increased.

Numerous studies [18,21,40,41,50,74,80,84] report an increase in the extent of sclerenchymatous tissues due to heavy metal loading. Contrary to the published data, in our experiment, we observed that the extent of the sclerenchyma bridges, which connect the vascular bundles to the epidermis, decreased. In the case of the sclerenchymatous tissues towards the adaxial surface, the decrease was significant.

Similarly to previous studies that examined the effects of heavy metal load on stomatal density, we observed that in the case of the BTF and SC Blossom lines, the number of the stomata increased both on the adaxial and abaxial epidermis (Table 5). While the number of stomata significantly increased on the adaxial epidermis in both lines, in the case of the abaxial epidermis, the increase in the number of stomata was statistically not verifiable. The increase in stomatal density was more particular in the BTF line. The stomatal density showed a growth parallel to the heavy metal load, which corresponds with the relevant literature [21,37,40,74,76,77,79,80].

During the examination period, the size of the stomata increased in the case of the Blossom and STM lines, both on the adaxial and abaxial epidermis, but the difference was not significant. However, in the case of the BTF line, a decrease in the size of the stomata was observed in the examined time interval, but the difference was not significant. Based on the work of previous authors [26,36,37,39,40], it can be established that a decrease in stoma size, in the case of the BFT line, indicates its exceptional adaptation ability.

As a result of the examination of the costal and intercostal rows, we can establish that their extent did not show a significant difference between any of the lines. However, in the case of the BTF and SC Blossom lines, a slight increase in size was noticeable on the adaxial epidermis in terms of both examined parameters. In the case of the BTF line, the number of cell rows in the costal region slightly increased on both the adaxial and abaxial epidermis (Table 5). We also detected an increase in the number of the cell rows building the intercostal zones in the BFT line, which is caused by the slight decrease in the size of the vascularity and the veins. This proves that the abiotic stress caused by toxic elements has fewer negative effects on this BFT line and that this line has better adaptational ability [24,25].

As for the STM and SC Blossom lines, the size reduction in the vascular bundles was less pronounced than in the case of the BTF line, where the decrease in this parameter showed a significant difference in all 3 years.

In all lines we examined, the increase in the phytoliths on the adaxial epidermis was easily detectable compared to the value from the first year. However, there was no significant difference between the years, except for the evolution of the number of phytoliths found on the adaxial epidermis of the BTF line. An interesting observation is that in the case of the STM line, in the second examination year, we noticed a sudden increase in the number of phytoliths. However, the results of the third year fell significantly short and even fell below the results of the first year. Phytoliths are also able to bind toxic elements, neutralizing their harmful effects and thus mitigating their negative effects, e.g., on photosynthetic processes. The above proves the good adaptability of the BTF line to heavy metal loads. Similarly to many authors [57,69,70,71,72], we concluded that the increased number of phytoliths due to abiotic stress is also a good indicator when examining the plant’s adaptational ability to heavy metal stress.

Overall, we can state that similarly to our previous observations [18], our present examination showed similar tendencies regarding the examined parameters. We can establish that the abiotic stress (whether it is drought, temperature, or even heavy metal/toxic element stress) significantly affects the parameters of the plants, in our case the leaf microanatomical parameters of the *Arundo donax* L. lines.

We can determine that out of the Arundo lines we examined, the BTF line showed good microanatomical adaptation abilities on soil contaminated with heavy metals. With this characteristic and its ability to produce a large amount of biomass, this line can be utilized for long-lasting phytoremediation of soils moderately contaminated with toxic elements. Its adaptational abilities are already shown at the tissue level; therefore, its physiological processes are less likely to be damaged than in the case of other examined lines (STM, SC Blossom), which showed weaker adaptational abilities.

## 3. Materials and Methods

### 3.1. Designation of the Experimental Site, Soil Tests

The experimental site was located in the Lovász-zug suburban area (47°29′05″ N, 21°35′40″ E, appr. 106 m above Baltic sea level) of Debrecen city (Hungary). Formerly, a sewage settling pond system was operated here as a secondary biological purification unit [85,86]. The sewage settling ponds were recultivated in 2013, and municipal sewage sediment (abbreviated as MSS) located at a 70–110 cm depth was covered with a soil layer [87,88]. Our former sampling and chemical analysis of the MSS [87] revealed that this wastewater solid is contaminated with potentially toxic elements (abbreviated as PTEs), since the measured values were: As–31.0, Ba–596, Cd–1.23, Cr–1142, Cu–198, Mn–520, Ni–62.8, Pb–278, and Zn–978 mg kg^−1^ on a dry-matter basis.

The genetic type of the cover soil used for recultivation could not be determined. In the surroundings of former sewage settling ponds, the prevailing soil type is chernozem in arable areas. To reveal and investigate the basic characteristics of the soil covering the MSS, an exploratory soil sampling was carried out in Debrecen Lovász-zug on 5 October 2017. At geographical point 47°29′01.17″ N, 21°35′46.17″ E from an area of 2.5 m × 2.5 m, 10 soil samples were taken randomly from a depth of 0–30 cm with Edelman augers (Royal Eijkelkamp, Giesbeek, The Netherlands). Approximately 15 kg of soil samples were collected and taken to the laboratory. After the removal of foreign substances, the soil was homogenized and spread on plastic plates in a thin layer. After 14 days of regular rotating and drying at room temperature, the air-dried samples were passed through a 2 mm sieve. From the thoroughly mixed air-dried soil, four mixed composite samples were taken. Each sample (with a mass of 70–110 g) consisted of 25 subsamples taken from different locations of two-finger-thick air-dried soil spread on plastic plates.

To reveal the concentrations of the PTEs in the topsoil of the experimental site, three sampling centers were designated. Sampling centers 1, 2, and 3 had geographic coordinates 47°29′3.87″ N, 21°35′44.07″ E; 47°29′3.95″ N, 21°35′45.03″ E; and 47°29′4.03″ N, 21°35′45.99″ E, respectively. A circle with a radius of 10 m was drawn from each sampling center with the help of a cord, and this was divided into 4 quarters marked as A, B, C, and D (Figure 4). Along a given radius, 2.5 m apart, four times two soil subsamples were taken from 0–25 cm depth with standard gouge augers (Royal Eijkelkamp, The Netherlands). After moving 15°, the samplings were repeated counterclockwise until a position of 90° was reached. In this way, 32 subsamples were collected from one sampling quarter with a total wet weight of 1000–1100 g. Sampling was repeated at segments A, B, C, and D. At all 3 sampling locations, soil sampling was conducted by 10 May 2018.

Soil samples were prepared, dried, and sieved, and air-dried soil was sampled as described above. One composite sample arose from the combination of 25 air-dried subsamples and had a 25–70 g weight on average.

### 3.2. Open-Field Experiment with Giant Reed

An open-field microplot experiment was set up with 3 ecotypes of giant reed (*Arundo donax* L.). The “STM” line, adapted to central European climatic conditions, was selected from a plant grown by Mr. Gyula Baja in the vicinity of Újszentmargita, Hungary. The “BFT” line was selected from a plant collected by Mr. Dr. László Márton in the state of Indiana, which is the northernmost giant reed-growing area of the United States. The “SC Blossom” line is from a plant collected by Mr. Dr. László Márton in the southeastern part of the USA (South Carolina, Columbia, Blossom Street, Congaree River bank) [89,90]. The STM line was presumed to be salt-tolerant, the BFT line was presumed to be cold-tolerant, and the SC Blossom line was presumed to produce a gigantic biomass yield. Cells of the above lines were callus micropropagated following the patent of Márton and Czakó [91] in the laboratory of Arundo Bioenergy Ltd. (Budapest, Hungary). Five-month-old bare-root plantlets were transferred to seedling soil and were acclimatized and grown in a greenhouse nursery for 2 months. Pre-grown giant reed plantlets were then planted in soil at the Lovász-zug experimental site by 24 May 2018. From line “STM” at Location 1, 64 plantlets, as well as 100 plantlets from lines “BFT” and “SC Blossom” at Location 2 or 3, were planted in an 8 m × 8 m or 10 m × 10 m square mesh, respectively. All the plants were one meter apart, and the distance between the edges of the 3 plots was 10 m (Figure 5). The geographical coordinates of the centers of the plots (Location 1, 2, or 3) are presented above as soil sampling centers.

### 3.3. Plant Sampling during the Experiment

The first sampling of giant reed leaves was conducted 16 weeks after planting, on 13 September 2018. From an experimental plot with a given Arundo line, 25 leaves were collected from segments A, B, C, and D (Figure 2). Twenty-five plants (not located at the edges of parcels) were randomly chosen from each sampling quarter, and the first fully developed leaves (having the 3rd position from the top of plants) were collected. The total fresh weight of 25 sampled leaves averaged 44 g. In this way, from a given Arundo line, 100 leaves were collected altogether with 4 replicate. Following the same protocol, a second sampling of giant reed leaves was conducted by 17 October 2019, and the third sampling was performed on 25 June 2020, when the plants were 73 and 113 weeks old. On 17 October 2019, the average fresh weight of 25 sampled leaves was 66 g, while On 25 June 2020, the average fresh weight was 96 g from one sampling quarter.

Immediately after sampling, leaves were thoroughly washed in flowing tap water in the laboratory. The tap water was rinsed from the samples in two-times-changed distilled water. Samples were dried until constant loss of weight in a drying oven (Mytron, Ltd., Heilbad Heiligenstadt, Germany) at 70 °C for 10 h. Dry samples were ground to particles < 1 mm in a ZEM 200 type an ultra-centrifugal mill (Retsch Ltd., Haan NRW, Germany).

The leaf samples for the microanatomical examinations were collected on 13 September 2018, 24 October 2019, and 28 September 2020. For the examinations, intact, healthy, mature leaves (from the middle, widest part of the 3rd leaf from the top) were collected. Three leaves from each sampling quarter, with twelve leaves altogether from one experimental plot, were collected at all sampling dates. The collected leaves were preserved in Strasburger–Flemming’s preservative solution (a mixture of 96% ethanol, 99.5% glycerol, and distilled water, 1:1:1 v v^−1^ ratio, [92]) until sectioning and preparation of the epidermis imprints.

### 3.4. Soil Samplings at the End of the Experiment

At the end of the experiment, on 8 June 2021, soil layer samples were taken from depths of 0–30 and 30–60 cm with Edelman augers (Royal Eijkelkamp, Giesbeek, The Netherlands). The location of the drilling was in the centers of plot quarters A, B, C, or D (Figure 2). The cleaning of soil samples from plant residues, drying and sieving, and sampling of air-dried soil was performed as described above. One composite sample (530–780 g) arose from combining 25 air-dried subsamples.

### 3.5. Element Analysis of Soil and Plant Samples

To determine the “pseudo-total” element content of the soil, the Hungarian Standard MSZ 21470-50 2006 was followed with slight modifications. The sample preparation procedure and the microwave digestion of 0.5 g of soil (<0.1 mm) in cc. HNO_3_ and in cc. H_2_O_2_ (3:1 v v^−1^) solutions are described in detail elsewhere [87,88].

To determine the soluble (“plant-available”) element content of the soil, the MSZ 20135 [93] was followed. The sample preparation procedure and the extraction of 0.5 g soil (<0.1 mm) with Lakanen–Erviö (LE) solution (0.02 M H_4_-EDTA in 0.5 M ammonium acetate buffer and 0.5 M acetic acid, pH 4.65; Lakanen and Erviö, 1971) is described in published articles [87,88].

From the prepared (dried and ground to particles < 0.1 mm) plant samples, 0.5 g was loaded into the pressure-proof bombs of the microwave digester (Milestone Ethos Plus, Sorisole BG, Italy). To all samples, 5 mL of distilled cc. HNO_3_ and 3 mL 30% (v v^−1^) H_2_O_2_ (Scharlau, Barcelona, Spain) were added [87].

Elemental analysis of all soil or plant samples was conducted with the inductively coupled plasma optical emission spectrometry (ICP-OES) technique, applied on an iCAP 7000 spectrophotometer (Thermo Fischer Scientific, Waltham, MA, USA). For the calibration, a multielement standard solution (n = 2) was applied. All element analyses were performed with 4 replicates.

### 3.6. Microanatomical Investigations of Leaves

Epidermis imprints and cross sections were made from leaf samples following the methods of Hilu and Randall [94], Gardner et al. [95], and Elagöz et al. [96]. Imprints were made from the adaxial and abaxial surfaces of leaves using clear nail polish. After drying the nail polish, imprints were examined under a BX51-type Olympus light microscope (Olympus BioSystems, Munich, Germany) [74,97]. The following micromorphometric parameters were examined: number of rows in the costal zone (no.), number of phytoliths (no./mm^2^), number of rows in the intercostal zone (no.), stomatal density (frequency of stomatal complexes (no./mm^2^), length of stomatal complexes (µm), and width of stomatal complexes (µm).

Leaf cross sections were prepared using razor blades following the method of Sass [98], and the examination of the cross sections was performed by using the above microscope. Preparations were colored with a 0.2% aqueous solution of toluidine blue (Merck KGaA, Darmstadt, Germany). The following micromorphometric parameters were examined: leaf lamina thickness (µm), adaxial and abaxial epidermis thickness (µm), leaf mesophyll thickness, vascular bundle width (µm), vascular bundle height (µm), vascular bundle area with bundle sheath (µm^2^), number of bulliform cells (no.), width of bulliform cells (µm), length of bulliform cells (µm), and thickness of adaxial and abaxial sclerenchyma bundles (µm).

A VSI RZ302 3M CMOS camera was used to prepare digital recordings, and a VSI RZ302 measuring program was applied to measure the above-mentioned micromorphometric parameters. Cross sections and epidermal imprints were digitally archived at 4 × 10, 10 × 10, and 10 × 20 magnifications. All investigated parameters were measured in 15–20 repetitions per treatment (for 3 Arundo lines and 3 examination years), and the obtained data were averaged.

### 3.7. Bioconcentration Factor (BCF)

The bioconcentration factor (BCF) [61] was calculated by dividing the given concentration of a PTE in an Italian reed leaf by the “pseudo-total” or “plant-available” given concentration of a PTE in soil.

### 3.8. Statistical Analysis of Data

The statistical analysis of experimental data was conducted with IBM SPSS Statistics 26.0 software using analysis of a variance (ANOVA) followed by treatment comparison using Tukey’s b-test. Correlation coefficients (r) between leaf and soil PTE concentrations were calculated according to the formula used by Microsoft Excel Office 16, and also with IBM SPSS Statistics 26.0 software using a two-tailed Pearson’s correlation.

## 4. Conclusions

As a result of our examination, we can establish that out of the three *Arundo donax* L. lines (BFT, STM, SC Blossom) we examined, the BTF line can be recommended for use in field practice for the phytoremediation of areas that are heavily contaminated with toxic elements. The adaptational abilities of the BTF line to toxic elements were manifested at the tissue level. As a result of the adaptation, the altered microanatomical properties (higher stoma number, smaller stoma size, more bulliform cells, extensive sclerenchymatous tissue, more phytoliths, vascular bundles with smaller diameters) defend the physiological processes of the plants. In this way, the plants can produce high-volume biomass, even in soil that is polluted with toxic elements. With regular harvesting of the aboveground biomass, reasonable amounts of toxic elements can be removed from the contaminated soil.

## Figures and Tables

**Figure 1 plants-13-00740-f001:**
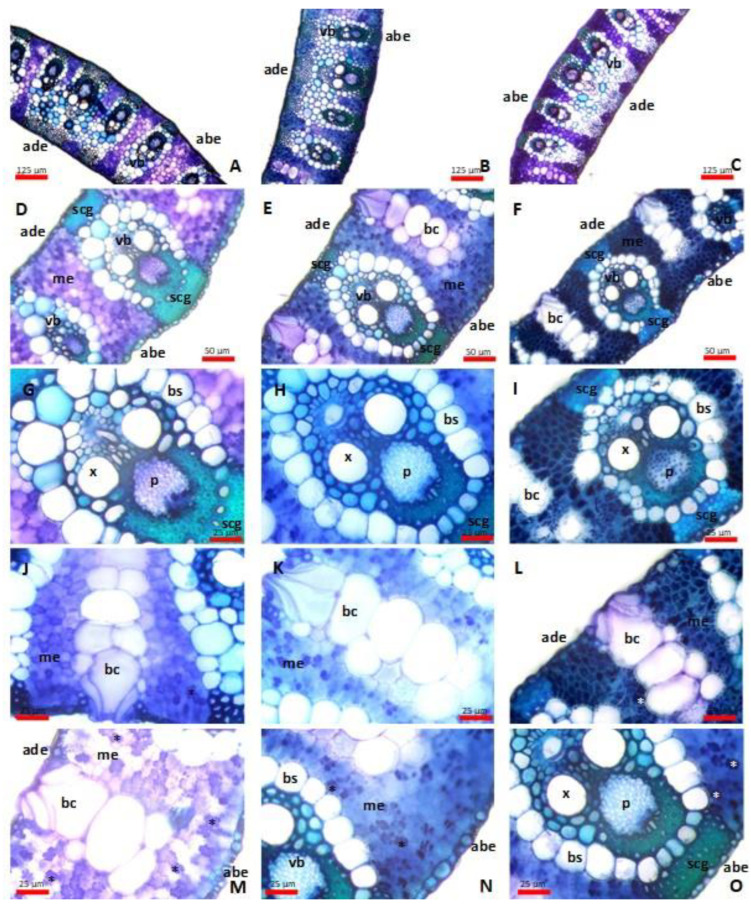
Transverse section of leaf laminae of *Arundo donax* L. (**A**–**C**) Median and lateral vascular bundles—BFT line, 2018 (4×); SC Blossom line, 2018 (4×) scale bars: 125 μm; STM line, 2018 (4×). (**D**–**F**) Leaf mesophyll—BFT line, 2020 (10×); SC Blossom line, 2018 (10×); STM line, 2020 (10×) scale bars: 50 μm. (**G**–**I**) Second-order vascular bundles—BFT line, 2020 (20×); SC Blossom line, 2018 (20×); STM line, 2020 (20×). (**J**–**L**) Bulliform cells—BFT line, 2018 (20×); SC Blossom line, 2018 (20×); STM line, 2020 (20×). (**M**–**O**) Phenolic compounds (asterisks) in the mesophyll cells—BFT line, 2018 (20×); SC Blossom line, 2018 (20×); SC Blossom line, 2018 (20×) scale bars: 25 μm. abe: abaxial epidermis, ade: adaxial epidermis, bc: bulliform cells, bs: bundle sheath, vb: vascular bundle, me: mesophyll, scg: sclerenchyma girder, x: xylem, p: phloem. Scale bars: 10 μm.

**Figure 2 plants-13-00740-f002:**
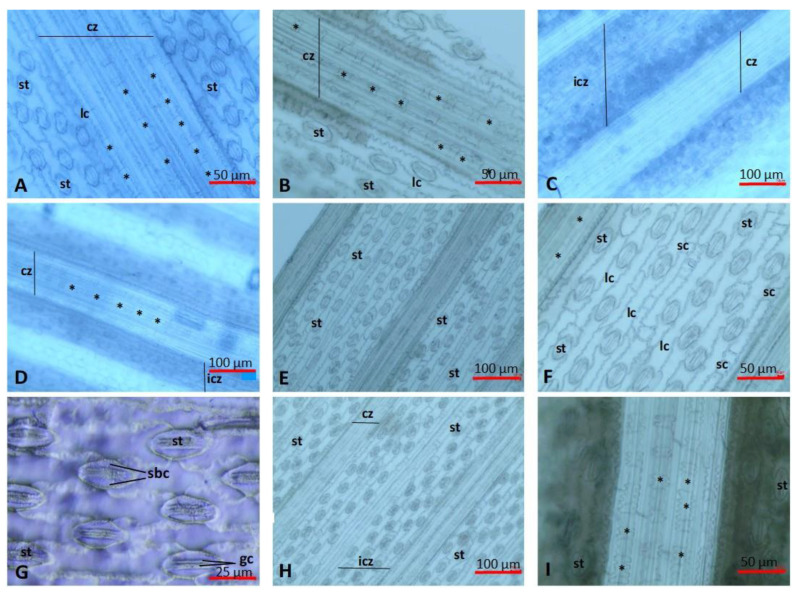
Leaf lamina epidermis of *Arundo donax* L. (**A**) SC Blossom line, adaxial epidermis, 2018 (20×). (**B**) SC Blossom line, adaxial epidermis, 2021 (20×). (**C**) SC Blossom line, abaxial epidermis, 2018 (20×). (**D**) STM line, adaxial epidermis, 2018 (10×). (**E**) STM line, abaxial epidermis, 2018 (10×). (**F**) STM line, abaxial epidermis, 2018 (20×). (**G**) STM line, abaxial epidermis, 2020 (40×). (**H**) STM line, abaxial epidermis, 2019 (10×). (**I**) BFT line, abaxial epidermis, 2021 (20×). cz: costal zone, icz: intercostal zone, lc: long cells, sc: short cells, st: stomata, gc: guard cells, sbc: subsidiary cells, *: phytoliths. Scale bars: 10×—100 μm, 20×—50 μm, 40×—25 μm.

**Figure 3 plants-13-00740-f003:**
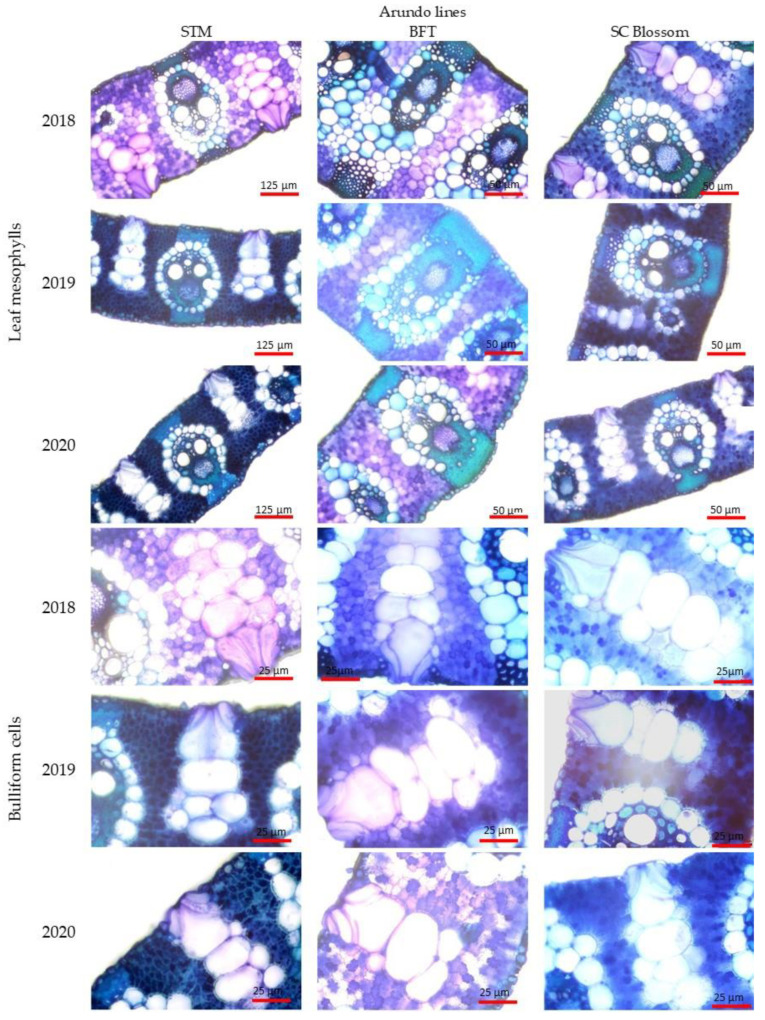
Evolution of the leaf lamina (4×—scale bars: 125 μm, 10×—scale bars: 50 μm) and bulliform cells (20×—scale bars: 25 μm) extent in the study years in the case of the individual *Arundo donax* L. lines examined open-field experiment, Debrecen Lovász-zug, Hungary, 2018, 2019, 2020). Scale bars: 10 μm.

**Figure 4 plants-13-00740-f004:**
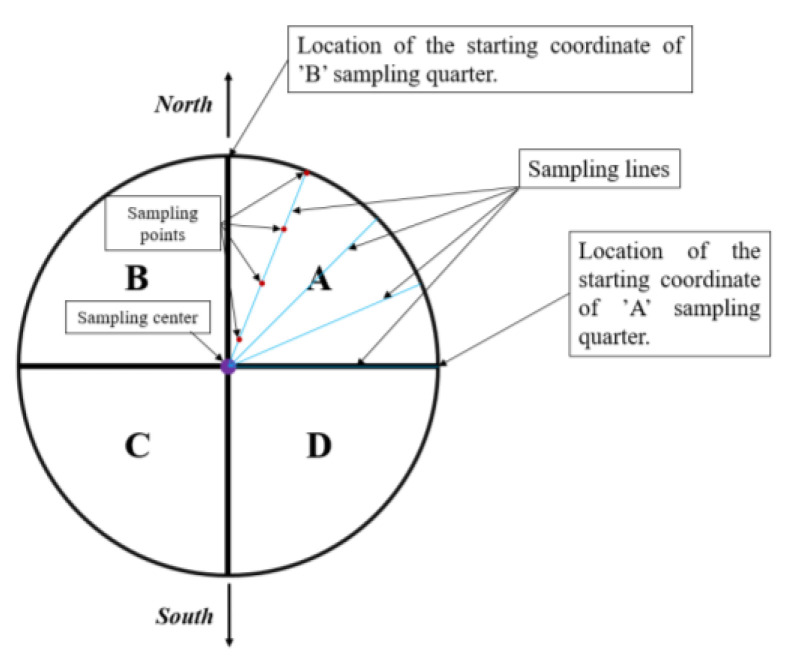
Scheme of soil sampling at the Lovász-zug experimental site (Debrecen, Hungary, 10 May 2018). The figure was compiled by Mr. Dr. Dávid Tőzsér (University of Debrecen, Hungary). Letters A, B, C, and D represent sampling quarters.

**Figure 5 plants-13-00740-f005:**

Scheme of the open-field microplot experiment setup with 3 lines of giant reed (*Arundo donax* L.) at the Lovász-zug experimental site (Debrecen, Hungary, 24 May 2018). Letters “A”, “B”, “C”, and “D” represent sampling quarters.

**Table 1 plants-13-00740-t001:** “Pseudo-total” concentrations of potentially toxic elements (PTEs) in the topsoil (0–25 cm) before starting the open-field experiment with Italian reed (*Arundo donax* L.) (Debrecen Lovász-zug, Hungary, 10 May 2018).

PTE µg g^−1^	Arundo Lines		
	STM (Location 1)	BFT (Location 2)	SC Blossom (Location 3)
As	9.85 ^a^	12.5 ^b^	13.9 ^c^
Ba	163 ^a^	434 ^c^	347 ^b^
Cd	0.301 ^a^	0.579 ^b^	0.731 ^c^
Cr	126 ^a^	310 ^b^	341 ^c^
Cu	39.3 ^a^	82.8 ^b^	88.4 ^c^
Mn	293 ^a^	411 ^b^	427 ^c^
Ni	21.9 ^a^	33.9 ^b^	37.1 ^c^
Pb	33.9 ^a^	92.5 ^b^	111 ^c^
Zn	156 ^a^	413 ^b^	435 ^c^

Data are means of four replicates. ANOVA Tukey’s b-test. Means within the lines followed by the same letter are not statistically significant at *p* < 0.05.

**Table 2 plants-13-00740-t002:** Concentrations of potentially toxic elements (PTEs) in the leaves of Italian reed (*Arundo donax* L.) (open-field experiment, Debrecen Lovász-zug, Hungary, 2018, 2019, 2020).

PTE µg g^−1^	Arundo Lines		
	STM (Location 1)	BFT (Location 2)	SC Blossom (Location 3)
13 September 2018			
As	0.094 ^b^	0.103 ^c^	0.065 ^a^
Ba	1.97 ^c^	1.82 ^b^	1.72 ^a^
Cd	0.329 ^b^	0.379 ^c^	0.270 ^a^
Cr	0.297 ^b^	0.564 ^c^	0.146 ^a^
Cu	13.2 ^b^	13.4 ^b^	11.4 ^a^
Mn	156 ^a^	179 ^b^	159 ^a^
Ni	0.611 ^b^	1.05 ^c^	0.468 ^a^
Pb	0.620 ^b^	1.28 ^c^	0.425 ^a^
Zn	170 ^b^	187 ^c^	147 ^a^
17 October 2019			
As	0.209 ^a^	0.272 ^b^	0.211 ^a^
Ba	2.25 ^b^	2.60 ^c^	1.96 ^a^
Cd	0.191 ^a^	0.297 ^b^	0.200 ^a^
Cr	0.070 ^a^	0.111 ^b^	0.079 ^a^
Cu	7.65 ^a^	10.8 ^b^	8.33 ^a^
Mn	111 ^a^	194 ^c^	128 ^b^
Ni	0.507 ^a^	1.05 ^c^	0.583 ^b^
Pb	0.369 ^a^	0.617 ^b^	0.392 ^a^
Zn	59.8 ^c^	117 ^b^	70.2 ^a^
25 June 2020			
As	0.214 ^a^	0.315 ^b^	0.224 ^a^
Ba	3.60 ^a^	5.89 ^b^	3.39 ^a^
Cd	0.501 ^b^	0.634 ^c^	0.467 ^a^
Cr	2.46 ^b^	2.94 ^c^	2.24 ^a^
Cu	8.39 ^a^	10.6 ^c^	9.88 ^b^
Mn	68.7 ^a^	107 ^c^	91.3 ^b^
Ni	3.17 ^a^	3.55 ^b^	3.05 ^a^
Pb	0.337 ^a^	0.624 ^c^	0.376 ^b^
Zn	83.9 ^a^	103 ^c^	92.7 ^b^

Data are means of four replicates. ANOVA Tukey’s b-test. Means within the lines followed by the same letter are not statistically significant at *p* < 0.05.

**Table 3 plants-13-00740-t003:** “Pseudo-total” and “plant-available” concentrations of potentially toxic elements (PTEs) in the topsoil (depth 0–30 cm) and subsoil (depth 30–60 cm) of the open-field experiment with Italian reed (*Arundo donax* L.) (open-field experiment, Debrecen Lovász-zug, Hungary, 8 July 2021).

PTE (mg kg^−1^)	Arundo Lines		
	STM (Location 1)	BFT (Location 2)	SC Blossom (Location 3)
“Pseudo-total” *	Soil depth 0–30 cm		
As	9.00 ^a^	10.1 ^b^	11.5 ^c^
Ba	158 ^a^	214 ^a^	293 ^b^
Cd	0.862 ^a^	1.28 ^b^	1.72 ^c^
Cr	133 ^a^	149 ^a^	267 ^b^
Cu	39.9 ^a^	44.2 ^b^	72.1 ^a^
Mn	403 ^a^	426 ^a^	425 ^a^
Ni	21.9 ^a^	26.2 ^a^	35.4 ^b^
Pb	28.3 ^a^	41.6 ^b^	67.1 ^c^
Zn	159 ^a^	291 ^a^	393 ^b^
	Soil depth 30–60 cm		
As	14.7 ^a^	15.3 ^b^	16.3 ^c^
Ba	212 ^a^	242 ^a^	348 ^b^
Cd	1.62 ^a^	1.76 ^a^	2.85 ^b^
Cr	209 ^a^	215 ^a^	412 ^b^
Cu	77.2 ^a^	80.4 ^a^	92.8 ^a^
Mn	429 ^a^	413 ^a^	456 ^a^
Ni	26.9 ^a^	29.0 ^a^	47.5 ^b^
Pb	46.7 ^a^	55.1 ^a^	92.6 ^b^
Zn	454 ^a^	427 ^a^	482 ^a^
“Plant available” **	Soil depth 0–30 cm		
As	1.11 ^a^	1.31 ^b^	1.51 ^c^
Ba	20.4 ^a^	19.9 ^a^	21.2 ^a^
Cd	0.603 ^a^	0.845 ^b^	1.14 ^c^
Cr	2.09 ^a^	2.05 ^a^	3.68 ^b^
Cu	19.3 ^a^	25.8 ^b^	41.3 ^b^
Mn	263 ^a^	275 ^a^	267 ^a^
Ni	5.06 ^a^	6.29 ^b^	7.98 ^c^
Pb	19.1 ^a^	29.0 ^a^	45.1 ^b^
Zn	92.7 ^a^	178 ^b^	240 ^c^
	Soil depth 30–60 cm		
As	1.50 ^a^	1.67 ^b^	1.77 ^c^
Ba	22.2 ^a^	23.9 ^b^	25.1 ^b^
Cd	1.11 ^a^	1.20 ^a^	1.92 ^b^
Cr	3.04 ^a^	2.90 ^a^	5.36 ^b^
Cu	37.8 ^a^	46.2 ^ab^	53.1 ^b^
Mn	275 ^a^	273 ^a^	287 ^a^
Ni	6.15 ^a^	7.17 ^a^	10.6 ^b^
Pb	31.4 ^a^	36.7 ^a^	62.1 ^b^
Zn	272 ^a^	256 ^b^	290 ^a^

* cc. HNO_3_ + cc. H_2_O_2_ soil extract, ** H_4_-EDTA in ammonium acetate buffer + acetic acid soil extract. Data are means of four replications. ANOVA Tukey’s b-test. Means within the lines followed by the same letter are not statistically significant at *p* < 0.05.

**Table 4 plants-13-00740-t004:** Anatomical characteristics of leaf tissues in different *Arundo donax* L. plant lines subjected to soil contamination from potentially toxic elements (PTEs) I.—Data from the leaf cross section (open-field experiment, Debrecen Lovász-zug, Hungary, 2018, 2019, 2020).

	2018	2019	2020
	Arundo Line: BFT		
Adaxial epidermis (µm)	13.88 ± 1.28 ^a^	13.06 ± 1.41 ^a^	13.69 ± 1.09 ^a^
Abaxial epidermis (µm)	15.83 ± 1.37 ^b^	14.50 ± 1.59 ^a^	14.96 ± 1.42 ^ab^
Leaf mesophyll thickness (µm)	421.29 ± 37.58 ^c^	335.07 ± 46.71 ^b^	256.18 ± 110.49 ^a^
Vascular bundle width (µm)	190.46 ± 11.19 ^c^	166.65 ± 13.63 ^b^	150.36 ± 61.25 ^a^
Vascular bundle height (µm)	264.53 ± 11.37 ^c^	232.44 ± 29.19 ^b^	183.84 ± 77.88 ^a^
Vascular bundle area + bundle sheath (µm^2^)	72,596.1 ± 6202.48 ^c^	54,967.57 ± 10,041.21 ^b^	33,619.79 ± 5461.28 ^a^
Bulliform cell number (no.)	4.2 ± 0 ^a^	4.35 ± 0.47 ^ab^	4.6 ± 0.6 ^b^
Bulliform cell height (µm)	109.21 ± 11.52 ^b^	91.69 ± 11.56 ^a^	91.29 ± 12.59 ^ab^
Bulliform cell width (µm)	314.22 ± 25.37 ^c^	268.18 ± 40.30 ^b^	212.47 ± 15.6 ^a^
Adaxial sclerenchyma (µm)	47.67 ± 4.88 ^b^	46.30 ± 7.49 ^b^	38.83 ± 9.0 ^a^
Abaxial sclerenchyma (µm)	42.27 ± 5.64 ^a^	49.47 ± 9.91 ^b^	40.29 ± 5.57 ^a^
	Arundo line: STM		
Adaxial epidermis (µm)	12.71 ± 1.1 ^a^	13.64 ± 1.54 ^a^	13.41 ± 0.87 ^a^
Abaxial epidermis (µm)	16.04 ± 1.96 ^b^	14.29 ± 0.99 ^a^	15.14 ± 1.67 ^ab^
Leaf mesophyll thickness (µm)	381.12 ± 24.11 ^c^	275.95 ± 21.98 ^b^	255.38 ± 27.5 ^a^
Vascular bundle width (µm)	186.84 ± 9.86 ^b^	148.04 ± 11.77 ^a^	146.79 ± 12.8 ^a^
Vascular bundle height (µm)	275.12 ± 18.31 ^b^	193.12 ± 16.11 ^a^	183.69 ± 25.03 ^a^
Vascular bundle area + bundle sheath (µm^2^)	67,349.27 ± 9236.71 ^b^	39,778.54 ± 5988.5 ^a^	38,102.77 ± 7324.76 ^a^
Bulliform cell number (no.)	3.8 ± 0.7 ^a^	4.5 ± 0.5 ^b^	4.6 ± 0.61 ^b^
Bulliform cell height (µm)	91.01 ± 12.06 ^a^	84.85 ± 8.63 ^a^	89.11 ± 10.14 ^a^
Bulliform cell width (µm)	273.71 ± 27.93 ^c^	231.96 ± 8.63 ^b^	203.28 ± 28.97 ^a^
Adaxial sclerenchyma (µm)	53.83 ± 13.62 ^b^	41.47 ± 6.33 ^a^	36.77 ± 5.57 ^a^
Abaxial sclerenchyma (µm)	42.29 ± 7.82 ^b^	36.86 ± 4.71 ^a^	34.41 ± 6.48 ^a^
	Arundo line: SC Blossom		
Adaxial epidermis (µm)	14.18 ± 1.04 ^b^	14.28 ± 1.64 ^b^	12.12 ± 1.16 ^a^
Abaxial epidermis (µm)	16.15 ± 1.32 ^a^	15.12 ± 1.75 ^a^	15.26 ± 1.04 ^a^
Leaf mesophyll thickness (µm)	363.54 ± 22.5 ^b^	294.59 ± 37.42 ^b^	299.79 ± 28.83 ^a^
Vascular bundle width (µm)	183.9311.75 ± ^b^	153.19 ± 12.93 ^a^	153.38 ± 8.77 ^a^
Vascular bundle height (µm)	246.93 ± 14.76 ^b^	200.25 ± 20.75 ^a^	200.38 ± 18.0 ^a^
Vascular bundle area + bundle sheath (µm^2^)	65,561.40 ± 9536.66 ^b^	41,548.18 ± 7317.44 ^a^	43,161.86 ± 4603 ^a^
Bulliform cell number (no.)	4.35 ± 0.57 ^a^	4.8 ± 0.6 ^b^	4.5 ± 0.49 ^ab^
Bulliform cell height (µm)	97.01 ± 11.1 ^b^	82.98 ± 8.31 ^a^	85.02 ± 12.43 ^a^
Bulliform cell width (µm)	274.18 ± 25.88 ^b^	222.92 ± 27.73 ^a^	237.67 ± 14.77 ^a^
Adaxial sclerenchyma (µm)	51.23 ± 7.94 ^b^	36.71 ± 7.42 ^a^	37.31 ± 4.34 ^a^
Abaxial sclerenchyma (µm)	39.90 ± 7.7 ^a^	36.42 ± 5.63 ^a^	40.97 ± 5.05 ^a^

Data are means of 15 replicates. ANOVA Tukey’s b-test. Means within the lines followed by the same letter are not statistically significant at *p* < 0.05.

**Table 5 plants-13-00740-t005:** Anatomical characteristics of leaf tissues in different *Arundo donax* L. lines plants subjected to soil contamination from potentially toxic elements (PTEs) II.—Data from the leaf epidermis (open-field experiment, Debrecen Lovász-zug, Hungary, 2018, 2019, 2020).

			2018	2019	2020
Arundo line: BFT	Adaxial epidermis	stomatal density (no./mm^2^)	196.8 ± 20.61 ^a^	257.6 ± 68.42 ^b^	310.4 ± 75.53 ^c^
		width of stomatal complexes (µm)	7.61 ± 0.76 ^a^	8.09 ± 2.16 ^a^	7.7 ± 2.13 ^a^
		length of stomatal complexes (µm)	31.13 ± 1.89 ^a^	34.08 ± 8.78 ^a^	32.84 ± 8.92 ^a^
		number of rows in the intercostal zone (no.)	4.4 ± 2.42 ^a^	4.8 ± 2.44 ^a^	4.8 ± 2.05 ^a^
		number of rows in the costal zone (no.)	4.6 ± 1.02 ^a^	5 ± 1.35 ^a^	5.6 ± 1.55 ^a^
		number of phytoliths (no./mm^2^)	206.8 ± 45.49 ^a^	246.2 ± 67.7 ^ab^	290 ± 77.87 ^b^
	Abaxial epidermis	stomatal density (no./mm^2^)	220.8 ± 20.61 ^a^	240.8 ± 89.05 ^a^	338.4 ± 86.51 ^b^
		width of stomatal complexes (µm)	8.51 ± 0.36 ^a^	8.16 ± 2.09 ^a^	6.8 ± 2.05 ^a^
		length of stomatal complexes (µm)	37.89 ± 1.05 ^a^	36.78 ± 9.24 ^a^	34.22 ± 9.31 ^a^
		number of rows in the intercostal zone (no.)	3.2 ± 1.47 ^a^	5 ± 2.25 ^a^	5.8 ± 2.64 ^a^
		number of rows in the costal zone (no.)	5.6 ± 0.8 ^a^	5.2 ± 1.42 ^a^	5 ± 1.64 ^a^
		number of phytoliths (no./mm^2^)	196.8 ± 35.57 ^a^	256.4 ± 79.13 ^a^	256.4 ± 71.78 ^a^
Arundo line: STM	Adaxial epidermis	stomatal density (no./mm^2^)	234.4 ± 19.03 ^a^	255.2 ± 8.19 ^a^	227.2 ± 20.77 ^a^
		width of stomatal complexes (µm)	7 ± 0.47 ^a^	7.75 ± 0.5 ^a^	8 ± 0.54 ^a^
		length of stomatal complexes (µm)	31.06 ± 1.34 ^a^	32.47 ± 0.81 ^a^	33.9 ± 0.99 ^a^
		number of rows in the intercostal zone (no.)	5.8 ± 2.64 ^a^	6 ± 2.76 ^a^	3.4 ± 1.02 ^a^
		number of rows in the costal zone (no.)	4.8 ± 0.4 ^a^	5 ± 0.89 ^a^	6 ± a.63 ^a^
		number of phytoliths (no./mm^2^)	186.6 ± 53.05 ^a^	246.6 ± 24.41 ^a^	289.6 ± 69.53 ^a^
	Abaxial epidermis	stomatal density (no./mm^2^)	267.2 ± 12.49 ^a^	314.4 ± 13.76 ^b^	199.2 ± 14.4 ^c^
		width of stomatal complexes (µm)	6.46 ± 0.32 ^a^	7.24 ± 0.73 ^a^	8.23 ± 0.74 ^a^
		length of stomatal complexes (µm)	33.14 ± 1.8 ^a^	33.48 ± 1.01 ^a^	34.04 ± 1.71 ^a^
		number of rows in the intercostal zone (no.)	5 ± 3.29 ^a^	5.4 ± 2.33 ^a^	4.8 ± 0.93 ^a^
		number of rows in the costal zone (no.)	4.4 ± 0.49 ^a^	4.8 ± 0.98 ^a^	5.4 ± 0.49 ^a^
		number of phytoliths (no./mm^2^)	229.8 ± 48.59 ^a^	313.4 ± 35.59 ^a^	219.8 ± 38.37 ^a^
Arundo line: SC Blossom	Adaxial epidermis	stomatal density (no./mm^2^)	178.4 ± 24.99 ^a^	246.4 ± 69.91 ^bc^	230.4 ± 59.50 ^b^
		width of stomatal complexes (µm)	6.68 ± 0.30 ^a^	5.52 ± 2.09 ^a^	8.03 ± 2.14 ^a^
		length of stomatal complexes (µm)	32.27 ± 1.07 ^a^	32.12 ± 8.75 ^a^	33.60 ± 9.11 ^a^
		number of rows in the intercostal zone no.)	5 ± 1.67 ^a^	6.8 ± 2.59 ^a^	5.8 ± 2.06 ^a^
		number of rows in the costal zone (no.)	4 ± 1.26 ^a^	4.2 ± 1.33 ^a^	4.6 ± 1.56 ^a^
		number of phytoliths (no./mm^2^)	200.2 ± 60.6 ^a^	206.6 ± 67.86 ^a^	266.4 ± 77.09 ^a^
	Abaxial epidermis	stomatal density (no./mm^2^)	292 ± 10.43 ^a^	309.6 ± 84.28 ^ab^	343.2 ± 99.97 ^b^
		width of stomatal complexes (µm)	7.08 ± 0.98 ^a^	6.19 ± 1.84 ^a^	7.17 ± 2.06 ^a^
		length of stomatal complexes (µm)	31.2 ± 0.92 ^a^	31.39 ± 8.86 ^a^	36.92 ± 9.5 ^a^
		number of rows in the intercostal zone (no.)	3.8 ± 1.46 ^a^	4.8 ± 2.46 ^a^	4 ± 2.31 ^a^
		number of rows in the costal zone (no.)	5.2 ± 0.4 ^a^	5.2 ± 1.37 ^a^	6.2 ± 1.61 ^a^
		number of phytoliths (no./mm^2^)	226.4 ± 58.83 ^a^	259.8 ± 81.82 ^a^	249.8 ± 69.84 ^a^

Data are means of five replicates. ANOVA Tukey’s b-test. Means within the lines followed by the same letter are not statistically significant at *p* < 0.05.

## Data Availability

Data is contained within the article and Appendix A.

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
