# Peer review of "Microanatomical Changes in the Leaves of Arundo donax (L.) Caused by Potentially Toxic Elements from Municipal Sewage Sediment"

_plants, 2024, doi:10.3390/plants13050740_

Round 1

Reviewer 1 Report

Comments and Suggestions for Authors

The regular paper of Tóth et al. depicts the effects of the potential toxic elements released in the soil due to the municipal sewage sediment on 3 micro propagated lines 13 (SC Blossom, BFT Indiana, and STM Hajdúsági) of giant reed (Arundo donax L.). This is considered as a mineral or heavy metal stress. The authors observe that the threshold tolerance of these giant reed lines differs between them based on the type of element and the location. Authors revealed that BFT Indiana shows the highest degree of tolerance and adaptations to toxic heavy metals as it accumulates the highest concentration of these toxic elements. This is confirmed by the assessment of the concentration of each element independent. It is mainly a comparative study to estimate the potential of three giant reed lines to support heavy metal stress (9 elements were checked). This was supported by the determination of each toxic element in both soil (before and after experiments) and the leaves based on different locations. They also investigated the effect of heavy metal on the anatomic section of leaves. Authors studied the effect of the stress on the cells size determined on the anatomical leaf sections for three consecutive years 2018-2020 and they found a decrease in the size or the number of certain cells such as bulliform cells and sclerenchymatous tissues. Hence, they displayed a decrease in the stomata size but an increase in the number and this is known as s sort of stress adaptation to abiotic stress such as drought and heat, and so on. Final they found that BFT is the best includer species of heavy metal in the leave and my serve as a potential robust phytoremediation species that farmer can use in the field to remove toxic elements before growing or sowing their large crops such as rice, wheat, soyabean etc.

Few simple points to check.

* Introduction is too long, which makes the manuscript not proportional

* Please all the words «stoma» through the manuscript text should be replaced by «stomata»

* The are minor English editing issues in the tense

* Line 615: Typos problem

Comments on the Quality of English Language

No comment

Reviewer 2 Report

Comments and Suggestions for Authors

The manuscript entitled “Microanatomical changes in the leaves of Arundo donax (L.) caused by potentially toxic elements from municipal sewage sediment” is devoted to an interesting and current topic, thanks to the fact that the studied plant is a prospective energy crop. The Authors investigated the uptake/accumulation of nine potentially toxic elements in the leaves of three giant reed lines (SC Blossom, BFT Indiana, and STM Hajdúsági), as well as their impact on anatomical changes during the tested period of three years. The results showed which line has the best adaptation ability and which is, therefore, suitable for growing in soils polluted with toxic elements. The data set is extensive, and the originality/novelty of this work is not questionable. However, there is still room for improvement.

Broad and specific comments

Abstract

In my opinion, there is no need to emphasize only two elements (Lines 16-17). So, I suggest deleting “(e.g. 413-465 16 mg/kg Zn, 126-345 mg/kg Cr)” and adding all examined elements in parentheses “(As, Ba, Cd, Cr, Cu, Mn, Ni, Pb, Zn)”. Further, in the next sentence, “The higher concentration…” the elements in parentheses are unnecessary (Lines 17-18).

Introduction

The Introduction chapter is too extensive. Results from previous research given in detail (Lines: 79-103, 125-135, 147-174) can be summarized and discussed later in the discussion.

Is the information given between lines 204 and 215 essential? Consider removing that part of the text. In case of such modification, the stated goals will be in the same paragraph, as is usual.

Results / Discussion

In the results, the authors have a lot of discussion elements, while the discussion is too concise. The Journal Plants allows the combination, so I suggest the authors merge the results and the discussion. Since the discussion is short, this correction shouldn't take much time. Further, if the authors replace some parts from the introduction, both sections will be highly improved and easier to follow.

The sizes of the bars are not visible in the images that show the anatomical features. Please correct Figures 1-3.

Materials and Methods

This section is well-written, clear, and informative. I have only one question. Why was the third sampling for microanatomical examinations done in September 2020, not June, when samples were taken for other analyses? On the other hand, why was the second sampling in June, not September or October? Probably the difference of a couple of months did not cause significant changes, but the fact is that the tested plants were of different ages.

References

The reference list is relevant and up to date. Check the references after correcting the introduction.

Best regards
